# Rapid wavefront shaping using an optical gradient acquisition

Sagi Monin ⓘ , Marina Alterman ⓘ & Anat Levin ⓘ ✉

Wavefront shaping systems enable deep tissue imaging by correcting scattering aberrations, but estimating optimal modulation correction is challenging, since it depends on the unknown tissue structures. Most current methods use slow coordinate descent algorithms, which sequentially scan all modulation parameters and query them independently, thus their complexity scales prohibitively with the number of parameters. We introduce a rapid wavefront shaping system, replacing coordinate descent with gradient descent optimization. To this end, our system acquires a gradient vector, which allows simultaneous update of all modulation parameters. We start with a non-invasive, guide-star-free score function to assess modulation quality and analytically derive its gradient with respect to all modulation parameters. Although the gradient depends on unknown tissue structure, we show it can be inferred from optical measurements. This enables fast, high-resolution wavefront correction with complexity independent of parameter count. We demonstrate the system's effectiveness in correcting aberrations in a coherent confocal microscope.

Optical imaging of tissue is challenging because cells and other tissue components scatter light. As a result, as light propagates deeper inside tissue, it becomes heavily aberrated, making it challenging to resolve clear images of deeper structures. Over the decades, several techniques, such as confocal microscopy[1] and optical coherence tomography (OCT)[2], have been developed to achieve deeper imaging by filtering out scattered photons and isolating ballistic photons. Despite this progress, these techniques are inherently limited to thin layers as the number of ballistic photons decays rapidly within scattering materials. To overcome this challenge, wavefront shaping was introduced. Instead of filtering out the scattered photons, wavefront shaping aims to measure the scattered light and invert the scattering process. Such corrections follow two major approaches: digital and optical.

Digital aberration correction methods attempt to illuminate the tissue using a set of wavefronts and measure the scattered light. This data is fitted with a parametric model to estimate the aberration and reconstruct the hidden target[3–21]. This approach has led to impressive results, but it is ultimately limited by signal-to-noise ratio (SNR). Even

with advanced gating strategies, light scattered by deeper tissue components is weak and can be lost in measurement noise.

An alternative class of techniques uses optical wavefront-shaping[22–27]. Such techniques use spatial light modulators (SLMs) to control the incoming and/or outgoing paths of the optical system. These SLMs reshape the incoming and/or outgoing wavefront in a way that is the inverse of the aberration it undergoes inside the tissue. This allows an incoming wavefront to focus to a diffraction-limited spot, and the light emitted from a single point within the tissue can be focused onto a single detector element. Unlike digital aberration corrections, the significant advantage of optical wavefront shaping is that the correction is physical; hence, all light photons emerging from a single target point can be brought into a single sensor point and measured with a significantly higher SNR.

Wavefront shaping ideas have found applications in a wide range of imaging modalities, including sound[24,28] and light, coherent imaging and OCT[29], and incoherent fluorescence imaging using single-photon[30–33] and multi-photon[22,34] excitation. This work showcases a simple aberration correction in a coherent, reflection mode confocal

Department of Electrical and Computer Engineering, Technion, Haifa, Israel. ✉e-mail: anat.levin.g@gmail.com

microscope. However, our framework also applies to other popular imaging schemes such as OCT and fluorescent microscopy.

The practical application of wavefront shaping is hindered by the complexity of determining the desired modulation correction. This wavefront correction differs between different tissue samples, and varies even within different regions of the same tissue sample. Earlier proof-of-concept demonstrations employed a validation camera behind the tissue to provide feedback for the algorithm[35–40]. Other approaches utilize a guide star[24,29,34,41–48], where scattering originates from a strong single point source within the tissue, allowing a wavefront sensor[43,48] to directly measure the scattered wavefront.

In the absence of such a guide-star, determining a wavefront shaping correction is a significantly more challenging task. Most approaches define a score function to evaluate the quality of the modulation based on the captured signal, and then optimize the SLM parameters to maximize this score. However, the optimal modulation depends on the unknown tissue structure. This structure can only be probed by projecting multiple modulation patterns and imaging their scattered output. Thus, most wavefront shaping optimization schemes employ coordinate descent (CD) approaches, where a single parameter of the modulation is varied at a time. Each optimization step involves selecting a new value for one parameter to improve the modulation score. These algorithms iterate sequentially through all degrees of freedom[30,32,34–40,49]. The main drawback of CD schemes is their time complexity, which scales with the number of free parameters in the modulation. However, for thick tissue, the modulation should ideally use all pixels on the SLM, usually in the megapixel range.

In this research, we derive a fast approach for estimating a wavefront-shaping correction in a non-invasive, guide-star free setting. In contrast to sequential CD approaches, our algorithm is capable of simultaneously updating all SLM parameters. To this end, we start by analytically differentiating the wavefront-shaping score. Although the gradient depends on the unknown tissue structure, we show that we can use optical computing to measure it. By simply capturing the back-scattered field, the gradient can be evaluated in closed-form. The dimensionality of the gradient vector is equivalent to the number of SLM parameters. With this gradient at hand, we can simultaneously update all SLM parameters, and we can transition from slow CD to fast gradient descent optimization.

Our gradient descent scheme is significantly faster than existing CD schemes and can also recover better modulations. CD methods typically restrict the number of SLM modes they optimize due to computational complexity and because the measurement of high-frequency modes is noise sensitive. However, in thick tissue where the scattering is wide, the optimal modulation requires a large number of degrees of freedom. In contrast, our approach can optimize a large number of modes without an increase in computational complexity.

We show that this higher number of modes allows us to find significantly better modulations.

## Results
### Imaging setup
In Fig. 1, we visualize a wavefront-shaping imaging setup. A laser beam illuminates a tissue sample via a microscope objective. A phase SLM in the illumination arm modulates the illumination pattern.

Coherent light back-scattered from the tissue target is collected by the same objective lens and reflected by a beam-splitter. A second phase SLM in the imaging arm modulates the returning wavefront. Lastly, the modulated light is measured by the front main camera.

Our setup incorporates a validation camera positioned behind the tissue sample. This camera serves to evaluate focusing quality and capture an undistorted reference image of the target. While earlier research demonstrations of wavefront-shaping utilized this camera to provide feedback to the algorithm, here we develop a non-invasive technique relying solely on feedback by the main (front) camera. It is crucial to note that the validation camera serves only for reference and does not provide any input to our algorithm.

To maximize the correctable area with a single modulation, the SLM should ideally be placed conjugate to the aberration[50], namely to a plane in the middle of the tissue. For simplicity, in our experimental setup, the SLMs are placed in the Fourier plane. To correct aberrations across a wide field of view with a single modulation pattern, we apply a tilt and shift operation to the Fourier pattern[51,52]. Critical to our derivation is the assumption that the two SLMs are placed such that they are conjugate to each other.

### Image formation model
We start by presenting a generalized image formation model for a dual SLM system, comprising an illumination SLM and an imaging SLM. Based on this model, we derive the specific model of our imaging system.

We denote the reflection matrix of the tissue as $\mathscr{R}$. This matrix describes propagation of light from the illumination SLM plane through the tissue and optics, and back to the imaging SLM. Thus, for every wavefront $\mathbf{u}^i$ placed on the illumination SLM, the resulting wavefront reaching the imaging SLM is modeled as a linear transformation by the reflection matrix:

$$\mathscr{R}\mathbf{u}^i. \tag{1}$$

Applying a modulation $\mathbf{u}^o$ to the imaging SLM, the wavefront reaching the camera is:

$$\mathscr{L}(\mathbf{u}^o \odot (\mathscr{R}\mathbf{u}^i)), \tag{2}$$

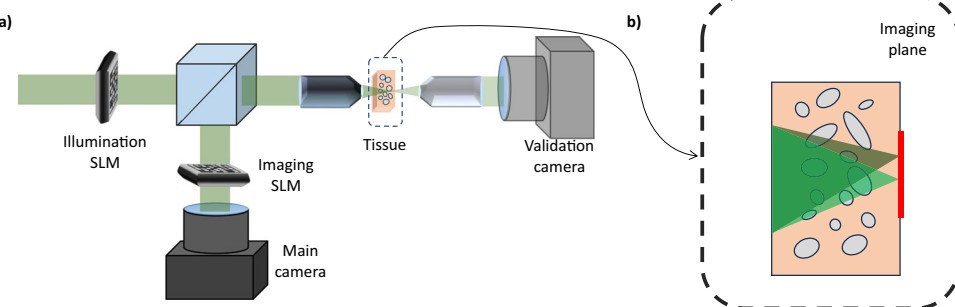

**Fig. 1 | System schematic. a** Schematic diagram of our system consisting of two SLMs. The first SLM modulates the illuminating laser light, while the second SLM modulates the reflected light. A validation camera provides reference images and confirms focusing on the desired target. **b** Our algorithm optimizes SLM phase to focus light on multiple adjacent points, by applying simple tilt-shift to the same modulation.

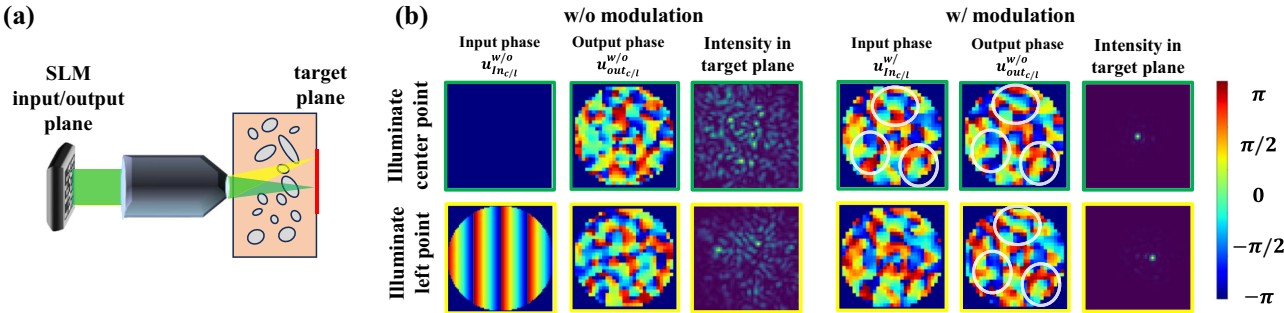

**Fig. 2 | Tilt-shift and time reversal effects. a** We show a simplified schematic of our system with two input wavefronts, green focusing into the center of the imaging plane and yellow focusing to a neighboring point to the left. **b** Simulation results showing the incoming phase on the SLM, the conjugate of the output phase after tissue reflection (when reaching the SLM plane), and the intensity at the target plane. With light modulation, strong correlations are observed between the incoming SLM phase and the output phase, as well as between output phases for different points (white circles indicate areas of strong correlation). Without aberration correction, correlation decreases rapidly.

where $\mathscr{L}$ denotes a linear mapping between the imaging SLM and the camera sensor that can be calibrated and described by a matrix, as it is independent of the tissue sample. $\odot$ represents element-wise multiplication between two vectors.

Our goal is to use the SLM modulation to image a small area inside the volume rather than a single point. For that, we denote the modulation parameters (the phase values of the SLM) as $\boldsymbol{\rho}^i$, $\boldsymbol{\rho}^o$. To direct this modulation to adjacent points within the scattering tissue, we apply tilt and shift to the SLM modulation[51,52]. We represent the tilt-shifted modulation directed towards volume point $\ell$ as $\boldsymbol{u}^\ell(\boldsymbol{\rho}^i)$. This notation implies that $\boldsymbol{\rho}^i$ are the parameters we can adjust, from which we derive multiple modulations $\boldsymbol{u}^\ell(\boldsymbol{\rho}^i)$ by applying fixed tilt-shift operations toward different points $\ell$. Similarly, $\boldsymbol{u}^\ell(\boldsymbol{\rho}^o)$ denotes a tilt-shifted modulation on the imaging SLM. With this tilt-shift, light emerging from different points $\ell$ in the volume arrives at the same position on the sensor, and is expressed as:

$$\mathscr{L}(\boldsymbol{u}^\ell(\boldsymbol{\rho}^o) \odot (\mathscr{R}\boldsymbol{u}^\ell(\boldsymbol{\rho}^i))). \tag{3}$$

The intensity at pixel $x$ of the camera is:

$$I(x) = |(\boldsymbol{\zeta}_x \odot \boldsymbol{u}^\ell(\boldsymbol{\rho}^o))^T \mathscr{R}\boldsymbol{u}^\ell(\boldsymbol{\rho}^i)|^2, \tag{4}$$

where $\boldsymbol{\zeta}_x$ is a row of the matrix $\mathscr{L}$ mapping the output of the SLM to a pixel $x$ on the camera.

In our system, $\mathscr{R}$ models the coherent propagation of a monochromatic laser beam. However, this formulation can be generalized to describe other forms of propagation, such as a path-length resolved reflection matrix describing the measurements of an OCT system. Also, $\mathscr{L}$ can represent various mappings between the imaging SLM plane to the camera sensor plane, depending on an optical setup of choice. In our system, we place the SLM in the Fourier plane; hence, $\mathscr{L}$ is a Fourier transform.

## Score function

We aim to develop an optimization framework that identifies the optimal wavefront modulation for focusing light inside the scattering tissue. To achieve this, we start by defining a score $\mathcal{S}(\boldsymbol{\rho})$ which quantifies a good modulation. We will then attempt to maximize this score to obtain a good modulation:

$$\hat{\boldsymbol{\rho}} = \arg\max_{\boldsymbol{\rho}} \mathcal{S}(\boldsymbol{\rho}). \tag{5}$$

To identify a good modulation in a non-invasive, guide-star free setting, we build upon the work of ref. 32 and recent digital correction approaches[9,10,14], and measure the confocal energy of the corrected area. For that, we apply the same modulation $\boldsymbol{\rho}^i = \boldsymbol{\rho}^o$ on both illumination and imaging SLMs, and denote it by a single parameter vector $\boldsymbol{\rho}$. We direct and collect light from points $\ell$ and measure the intensity at the central sensor pixel. In our system, the mapping between the SLM and the camera is a Fourier transform. Consequently, the central pixel corresponds to the DC component of the Fourier transform (i.e., $\boldsymbol{\zeta}_x$ is a uniform vector). Following Eq. (4), we can express the measured confocal energy as:

$$I(0) = |\boldsymbol{u}^\ell(\boldsymbol{\rho})^T \mathscr{R}\boldsymbol{u}^\ell(\boldsymbol{\rho})|^2. \tag{6}$$

We sum the confocal intensity over an area and measure the total intensity we can collect by a confocal scanning of the modulation tilted toward different points $\ell$ in an area of interest $\mathscr{A}$:

$$\mathcal{S}(\boldsymbol{\rho}) \equiv \sum_{\ell \in \mathscr{A}} |\boldsymbol{u}^\ell(\boldsymbol{\rho})^T \mathscr{R}\boldsymbol{u}^\ell(\boldsymbol{\rho})|^2. \tag{7}$$

To better illustrate this score, Fig. 2 visualizes the incident and reflected wavefronts, both with and without modulation, for two points within the target area. Initially, without modulation, these wavefronts exhibit minimal correlation. However, once we find a focusing modulation, the incoming and outgoing wavefronts demonstrate higher correlation. Additionally, we observe a strong memory effect correlation among wavefronts returning from different points.

We note that in ref. 32, an incoherent fluorescence signal was collected, and the confocal energy was measured at a single pixel, without attempting to average over an area. Since we use coherent light, we notice that if we only direct light to a single spot, a high confocal score at the sensor plane does not guarantee that the light has focused into a single spot inside the tissue due to various interference effects. To address this, we propose averaging the confocal score over a small area $\mathscr{A}$, which mitigates these interference effects. This approach aligns with recent coherent digital correction algorithms[9,10,14] that optimize over finite isoplanatic patches to maximize the diagonal elements of the reflection matrix, which is equivalent to the confocal area score. The supplementary file provides a short derivation justifying the score using local memory effect correlations. Fig. 3 in the supplementary material shows a simulation comparison of an input/output modulated vs unmodulated wavefronts. We investigate the impact of the target area size on focusing performance in our experimental section, and show that in practice, one modulation can only be used over a very small area due to the limited extent of memory effect correlations.

## Optical gradient acquisition

The main challenge in utilizing the score defined in Eq. (7) lies in its dependence on the tissue's reflection matrix, which is inherently linked to the unknown tissue structure. Since the tissue is treated as a black box, most previous approaches have employed CD optimization strategies[30,32,34–36,49]. In each iteration, only one parameter is varied (often in a Hadamard basis) and adjusted to improve the measured score. For invasive setups, the score is simply the intensity measured by a validation camera at a point behind the tissue. Given that modern SLMs have mega-pixel resolution, sequentially scanning each free parameter is an extremely time-consuming process. Consequently, researchers frequently reduce the SLM resolution and correct a significantly smaller number of modes. Below, we derive a scheme for simultaneously measuring the gradient with respect to all SLM parameters. This approach enables us to transition from CD schemes to a gradient descent scheme that updates all SLM parameters in each iteration.

We start by applying the chain rule and differentiate Eq. (7) with respect to the SLM wavefront $\boldsymbol{\rho}$:

$$\frac{\partial \mathcal{S}(\rho)}{\partial \rho} = 2\sum_\ell \underbrace{\left(u^\ell(\rho)^T \mathcal{R} u^\ell(\rho)\right)^*}_{(1)} \cdot \underbrace{\left(\mathcal{R} u^\ell(\rho) + (u^\ell(\rho)^T \mathcal{R})^T\right)}_{(2)} \odot \underbrace{\frac{\partial u^\ell(\rho)}{\partial \rho}}_{(3)}. \quad (8)$$

This gradient is a product of three terms. The first (1) is a complex scalar. The second (2) is a complex vector whose dimensionality is equivalent to that of the modulation. The third term (3), which differentiates the wavefront with respect to the parameters $\boldsymbol{\rho}$, can be computed explicitly. As we explain in the supplementary material, since $\boldsymbol{u}^\ell(\boldsymbol{\rho})$ simply shifts and multiplies the elements of $\boldsymbol{\rho}$ by complex tilt scalars, the derivative with respect to $\boldsymbol{\rho}$ is also a simple tilt-shift of the derivative with respect to $\boldsymbol{u}^\ell(\boldsymbol{\rho})$.

The challenge with the first two terms is that they involve the unknown reflection matrix $\mathcal{R}$. Nevertheless, we can leverage the optics to measure the gradient in Eq. (8) for any candidate modulation $\boldsymbol{\rho}$ of interest. We observe that the complex field $\mathcal{R}\boldsymbol{u}^\ell(\boldsymbol{\rho})$ is essentially equivalent to applying a modulation at the illumination arm only, and capturing a non-modulated speckle field. The complex field can be captured using a phase retrieval algorithm or with an interferometric imaging system, as detailed in the supplementary material. With this measurement, we can numerically compute the remaining terms of Eq. (8). For example, following the reciprocity principle, $\mathcal{R}$ is a symmetric matrix and $(\boldsymbol{u}^\ell(\boldsymbol{\rho})^T \mathcal{R})^T$ is equal to $\mathcal{R}\boldsymbol{u}^\ell(\boldsymbol{\rho})$. Similarly, the term (1) in Eq. (8) can be computed by multiplying $\mathcal{R}\boldsymbol{u}^\ell(\boldsymbol{\rho})$ with the known vector $\boldsymbol{u}^\ell(\boldsymbol{\rho})$.

By measuring the gradient of the modulation score, we can simultaneously update all entries of the SLM. This offers a substantial acceleration compared to previous approaches, which iterate through the entries and update only one degree of freedom at a time.

In our current implementation, we measured the fields $\mathcal{R}\boldsymbol{u}^\ell(\boldsymbol{\rho})$ using a phase diversity scheme. This involved applying five defocus wavefronts to the imaging SLM[53,54], capturing five speckle intensity images, and solving an optimization problem for recovering the phase[55]. We measured each of the $\ell$ points and summed them. In the supplementary material, we show that the gradient can in fact be imaged directly using a point-diffraction interferometry scheme[56,57], bypassing the optimization. Moreover, the summation across all points $\ell$ can be computed within a single exposure; hence, the entire gradient can be imaged with as little as three shots.

We note that the gradient formula in Eq. (8) assumes $\boldsymbol{\rho}$ is a complex vector. In practice, most SLMs only modify phase. In supplementary sec. 3.1, we provide a small adaptation of Eq. (8), which provides the derivative with respect to phase.

In the supplementary file we also explain that the gradient descent approach derived above resembles fast wavefront shaping algorithms based on time reversal and the power algorithm[28,31,58–61]. In fact, it is equivalent to a power iteration of the reflection matrix if one aims to maximize confocal intensity at a single spot rather than an area. However, as we show below, a single spot score does not result in good modulations. While power iterations are tied to an eigenvector calculation and may not be adjusted easily to optimize other things, defining a general score function and its gradient provides a more principled way to impose all sorts of desired properties on the solution.

The optical calculation of the gradient here also resembles the derivation in ref. 62. However, their system uses transmission mode imaging and thus requires two SLMs on two sides of the scattering medium.

## Experimental results

In our experiments, we used three types of targets. The first consisted of high-reflectance chrome-coated masks (Nanofilm), patterned using an in-house lithography process to create structures with a resolution of 2 μm. We placed these behind scattering layers composed of chicken breast tissue with a thickness of 130−240 μm or with a number of layers of parafilm[63]. The second target included polystyrene beads dispersed in agarose gel. Finally, we imaged inside an onion layer.

We start by showing convergence of our algorithm. In Fig. 3, we cover the chrome mask with 180 μm thick chicken breast, and demonstrate the algorithm's efficiency, converging to the desired spot in as few as ten iterations, compared to the thousands required by previous CD algorithms. We show an image of one of the points $\ell$ in the optimized area $\mathscr{A}$, with no modulation and when the optimized modulation is placed on both SLMs. Without modulation, we see a noisy speckle image, but with the modulation, we see a sharp spot whose intensity is about 20× higher. We also image the chrome mask directly using the validation camera. Without modulation, we see a wide speckle pattern, but with the optimized modulation on the illumination arm, all light propagates through the scattering layer and is focused into a single point on the chrome mask. These validation images ensure that the point we see in the main camera indeed corresponds to a point on the mask, and we are not achieving a point on the main camera by complicated interference effects inside the tissue. At the lower row, we show the progress of the phase mask during the iterations.

Next, we evaluate the effect of the area $\mathscr{A}$ over which the score is optimized in Fig. 4. If this area is small, we observe a sharp spot at the main camera, but the light does not actually focus into a spot inside the tissue, as we do not observe a spot at the validation camera. This indicates that the spot at the main camera results from interference of coherent light within the tissue. This problem does not occur with fluorescent wavefront shaping[32] since the incoherent emissions from nearby fluorescent points do not interfere. As we increase the size of the scanned area, requiring the same modulation to explain multiple points, such interference effects disappear, and we achieve a point in both cameras. However, as we increase the target area, the intensity of the focused spot at the main camera decays because the correction required by nearby points varies spatially and one modulation cannot explain a wide area. This is because memory effect correlation does not hold for very large ranges. Note that our score function recovers modulations using a non-invasive, guide-star free feedback.

Next, we show that our system can retrieve an image of an area by performing a confocal scan. In Fig. 5, we present imaging results for a few patterns printed on the chrome mask. We display images of a single point in the area through both the main and validation cameras. We also show a confocal scan of the entire area. Without modulation, this looks like a speckle pattern; however, with the modulation, the confocal scan reveals the shape of patterns printed on the chrome mask. Note how the modulation increases light by a factor of 12−100×. In Supplementary Fig. 7, we show the individual images captured for many points $\ell \in \mathscr{A}$ in both the main and validation cameras. The result

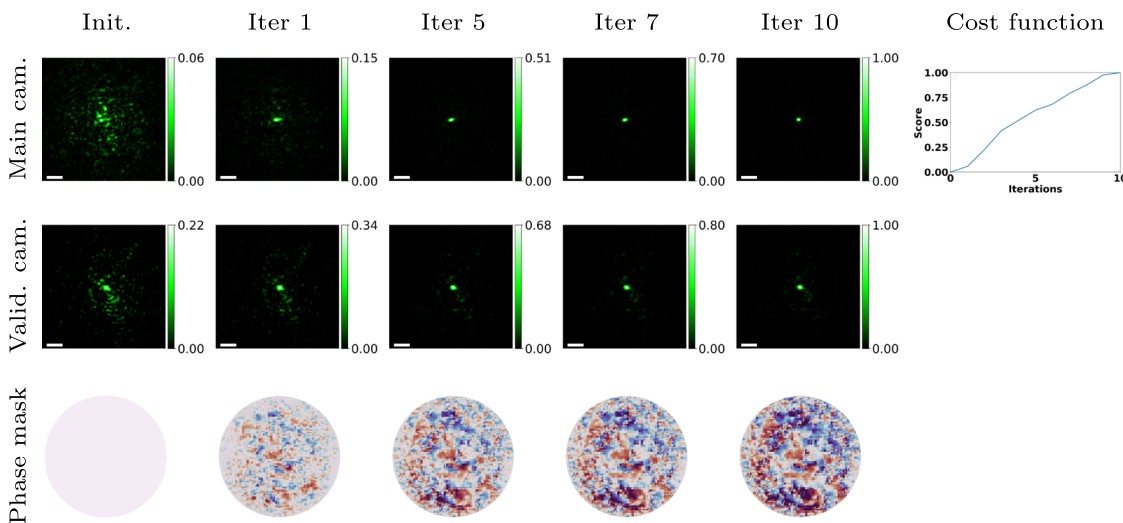

**Fig. 3 | Algorithm convergence.** We used our algorithm to image a glass mask partially covered with chrome, placed behind 180 μm thick chicken breast tissue as the scattering material. We show captured images on both the main and validation cameras before the algorithm is applied and at the end of each iteration. The first row demonstrates the algorithm's convergence over iterations on the main camera for one of the scanned points. The second row displays the convergence on the validation camera, demonstrating our ability to also focus light on the target plane. The final row depicts the evolution of the phase mask presented on both SLMs. The last column illustrates the improvement of the cost function across iterations. The scale bar is 4 μm.

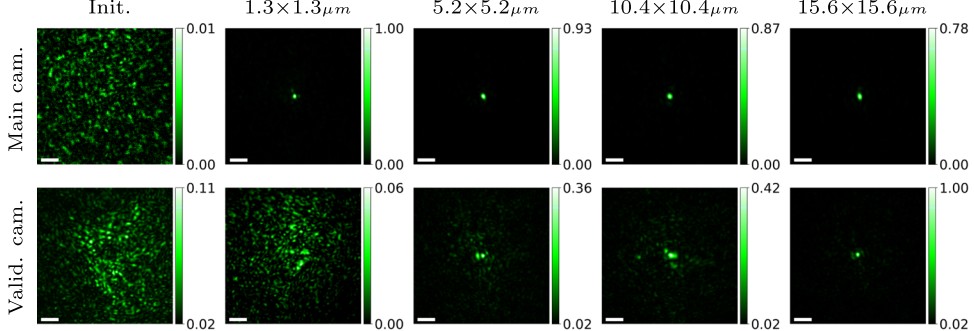

**Fig. 4 | Effect of target area size.** We evaluate how focusing quality depends on the size of the target area $\mathscr{A}$ over which the confocal score is computed. The experiment used a chrome-covered glass mask overlaid with 240 μm thick chicken breast tissue. Our results show that for single-spot illumination (second column), the algorithm successfully achieved a sharp focused spot on the main camera without actually focusing light to a spot inside the tissue, as evident by the image from the validation camera. As we expanded the size of the focus area, the algorithm demonstrated progressively improved ability to focus light inside the tissue, and the validation camera images were sharper. At the same time, increasing the scanned area reduces the intensity of the focused spot on the main camera because a single modulation cannot correct very large areas. The scale bar is 4 μm.

in the last row of Fig. 5 was larger than the range of memory effect correlation. To overcome this, we imaged multiple local corrections and stitched the patches together. In Supplementary Fig. 8, we show the different isoplanatic patches.

Next, we demonstrate that our algorithm can also image translucent targets with small variations in refractive indices. To do this, we disperse polystyrene beads in agarose gel. Polystyrene and agarose have refractive indices of approximately 1.598 and 1.334 under our 532 nm illumination, and thus correspond to the practical range of refractive indices of tissue components. The bead diameter is 0.5 μm, and the slab thickness is 1.3 mm. The left part of Fig. 6 illustrates the schematic of this target, where we used our algorithm to focus inside the volume and image a 3D sub-volume. We estimate that the optical depth (OD) of this sample is 2.5 by measuring the attenuation of ballistic light in the validation camera. In Fig. 6, we visualize the confocal scanning result. Our target area $\mathscr{A}$ was now a 3D sub-volume rather than a 2D patch, where beyond tilt-shift, we added in $\boldsymbol{u}^{t}(\boldsymbol{\rho}^{t})$ a quadratic phase to focus light at varying depths. In our results, we present a few $x-y$ cross sections of planes in the corrected sub-volume. In addition, we

display the maximum projection onto the three different axes, showing good agreement between our aberration-corrected results and the reference. While a standard confocal scan of such beads is very noisy, with our estimated modulation, we can largely increase its contrast and achieve a clear image of spots corresponding to the beads' positions. To obtain a reference image of the bead positions, we image beads at the further depth of the slab, closer to the validation camera, so we are able to see an aberration-free reference in the validation camera. For this reference image, we used a wide-field incoherent illumination. In Supplementary Figs. 5 and 6, we show additional results of 2D slices inside a beads dispersion, where we use beads of various sizes.

Utilizing our algorithm, in Fig. 7, we successfully imaged onion cells through a 130 μm thick slice of highly scattering onion[64]. Specifically, we visualized the intercellular boundary between two neighbouring cells. In Supplementary Fig. 9, we show a larger field of view of the onion under incoherent illumination. In addition, we show a confocal scan of an onion cell at a shallower layer in Supplementary Fig. 11, where scattering does not limit image quality. The aberration-corrected result in Fig. 7 matches this unaberrated structure.

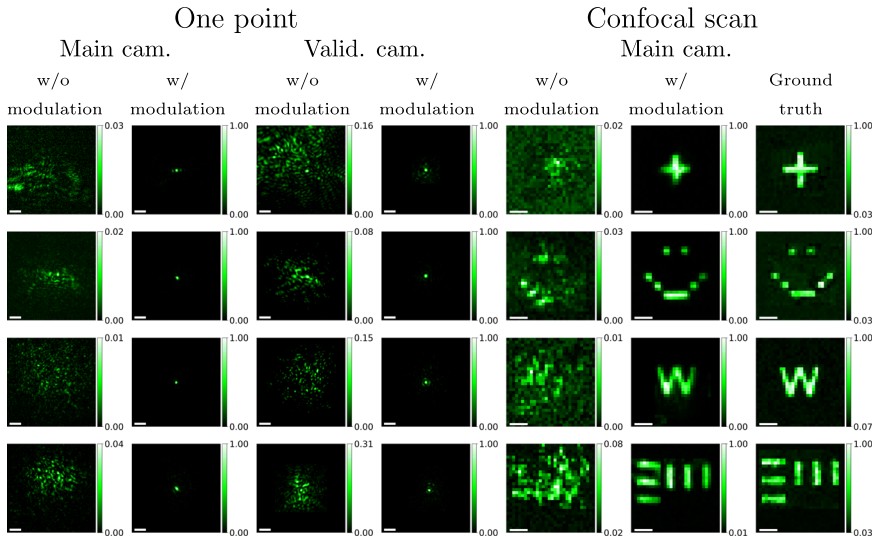

**Fig. 5 | Confocal imaging.** We perform confocal imaging of patterns printed on a chrome mask, through a scattering layer. Our algorithm successfully focuses light on the desired plane. Each row shows a different target. Columns 1–2: Main camera images of a single focal spot before and after optimization. Columns 3–4: Corresponding validation camera images. Columns 5–7: Confocal scanning results: Without aberration correction, with aberration correction, and reference scan of the target before applying scattering material, respectively. In the first three targets (top rows), the scattering material is chicken breast of thickness of 130, 150, and 200 μm, respectively. In the fourth target, the scattering material is two layers of parafilm. The scale bar is 4 μm.

The area of the result achieved in Fig. 7 was bigger than the range of the memory effect correlation and cannot be considered as a single isoplanatic patch. Hence, we imaged this target by applying multiple local corrections and stitching a few patches together. In Fig. 10 of the supplementary material, we show the different isoplanatic patches. In Fig. 8, we visualize the 3D structure of the onion by displaying corrections at three different depth slices. We show additional slices in Supplementary Fig. 12.

In the following paragraphs, we use again the chrome mask target to evaluate various components of our algorithm.

In contrast to CD optimization, whose time complexity scales with the number of modes corrected, the time complexity of our gradient descent approach does not depend on the number of corrected modes. In practice, to reduce scan time, CD schemes optimize for a relatively small number of modes. However, for thick tissue where the scattering is wide, we require a large number of modes to obtain a good correction. In Fig. 9, we demonstrate the impact of resolution on the quality of the correction. As shown, a modulation with more degrees of freedom allows us to focus more energy into a single spot.

Next, we explicitly compare our gradient descent scheme with a CD approach. Following refs. 32,37, we express the modulation as $\rho = \sum \phi_k H_k$, where $H_k$ are elements in a Hadamard basis. We scan the basis elements, adjusting the phase of one basis element at a time. Our algorithm clearly converges in significantly fewer iterations, as all parameters can be updated simultaneously. However, we argue that the resolution of CD schemes is not only limited by acquisition time but also by SNR. For high-frequency modes, varying $\phi_k$ results in a very small change to the score, potentially lower than the noise level. In Fig. 10, we show that the CD score cannot improve beyond a certain point. To illustrate that noise is the limiting factor, we run another CD scheme, but this time we averaged multiple shots for each basis element to reduce noise. This oversampling approach allowed us to update the high-frequency modes, leading to much better results. However, the oversampling is also 5× slower than the standard CD scheme and 125× slower than our gradient descent scheme. The modulation computed in Fig. 10 is a low-resolution one, consisting of only 256 parameters. For higher resolution modulations, the time difference between CD and gradient descent approaches would be even larger.

Finally, in Supplementary Fig. 13, we further compare our method with the CLASS algorithm[5], a representative digital correction technique[7,8,13,14]. Digital approaches such as these measure the scattered wavefront and numerically fit it with a parametric model that describes the underlying aberration and the hidden target. Because the measured scattered fields are typically very noisy, the reconstruction quality is ultimately limited by the SNR of the recorded data. In contrast, performing the correction optically–by applying the estimated modulation directly on the SLM–enhances the effective SNR of subsequent measurements, leading to cleaner reconstructions and improved correction fidelity.

## Discussion

The challenge in computing a wavefront shaping modulation lies in its dependence on the unknown tissue structure. Consequently, most previous optimization strategies relied on slow CD strategies, which sequentially scan and query the modulation parameters one at a time. Our contribution lies in deriving an analytic method to compute the gradient of the wavefront shaping score directly from the scattered wavefront. Crucially, we can compute the gradient in closed-form, despite the unknown tissue structure. This enables a transition from CD to gradient descent optimization, allowing simultaneous updates to all SLM parameters. The resulting optimization strategy is orders of magnitude faster than previous CD approaches. Moreover, our approach facilitates the estimation of high-resolution modulations, leading to superior corrections when stronger scattering is present.

We demonstrated the applicability of our gradient acquisition framework using a simple reflection-mode confocal microscope setup. This confocal system is inherently limited due to weak back-reflection of biological samples and the mixing of reflections from different depth layers, resulting in poor depth sectioning. To enhance the utility of wavefront shaping corrections in real tissue imaging, our framework can be integrated into OCT or fluorescence microscope systems. OCT enables imaging deep within tissue volumes by isolating content at different depths through path-length filtering. However, in highly scattering tissues, OCT filtering is significantly affected by aberrations, which can be effectively corrected using a wavefront shaping system. Since OCT relies on coherent light, the optimization framework presented in this paper is directly applicable to OCT and can facilitate the

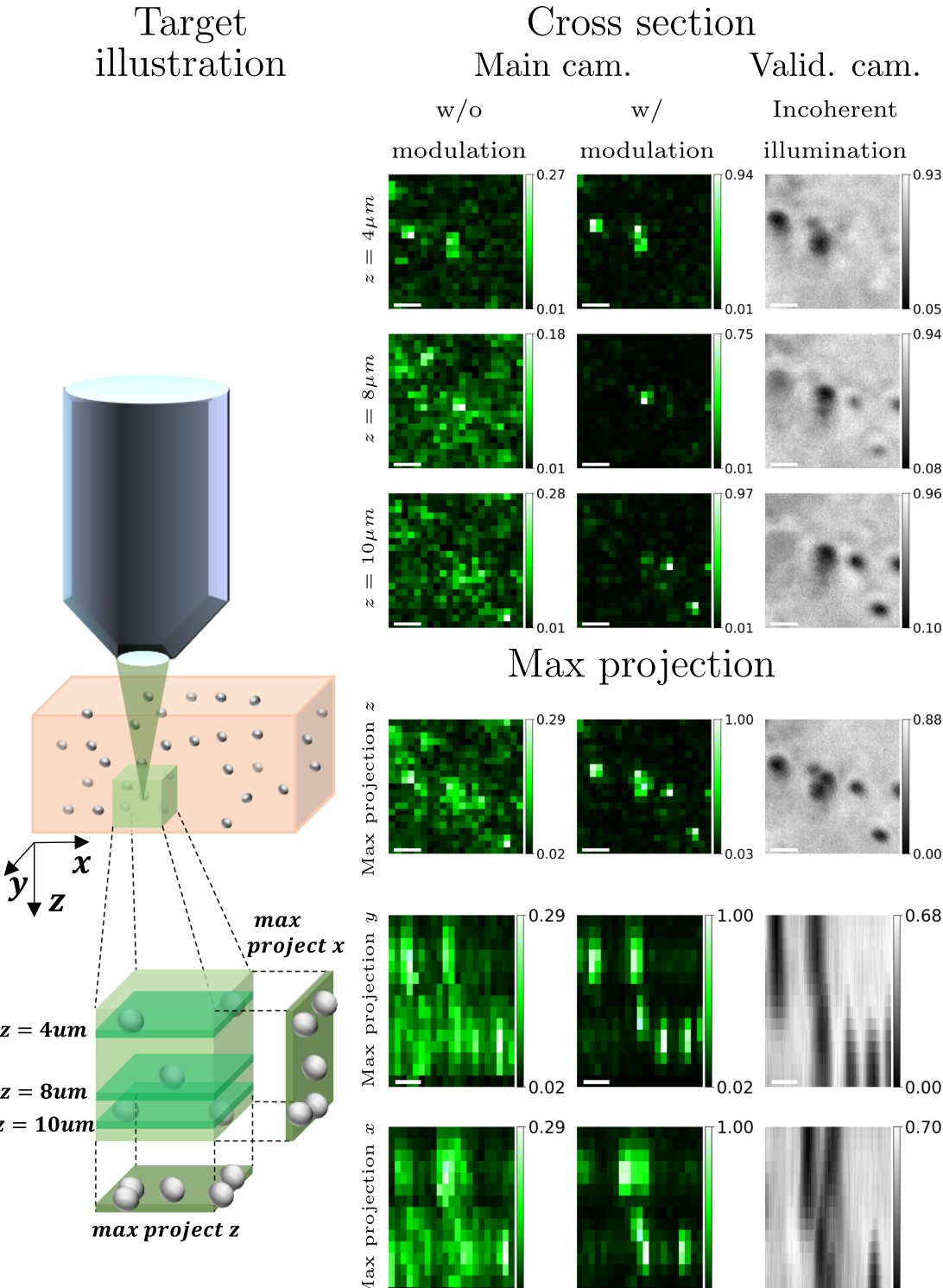

**Fig. 6 | Confocal of 3D volume.** Our algorithm was used to image a scattering dispersion of 0.5 µm diameter polystyrene beads in agarose gel. The first column depicts a schematic view of the target, where we image a small volume of size 10.4 × 10.4 × 14 µm. The green bars inside the volume depict different cross sections we present to the right. In addition, we present a maximum projection to each of the three axes. Columns 2–3: Confocal scanning results: with and without aberration correction. Column 4: A reference image of the target captured from the validation camera behind the target, under wide-field incoherent illumination. Rows 1–3 show three *xy* cross sections at different depths 4, 8, 10 µm. Rows 4–6 present the maximum projections onto each of the axes. The scale bar on confocal images is 2 µm.

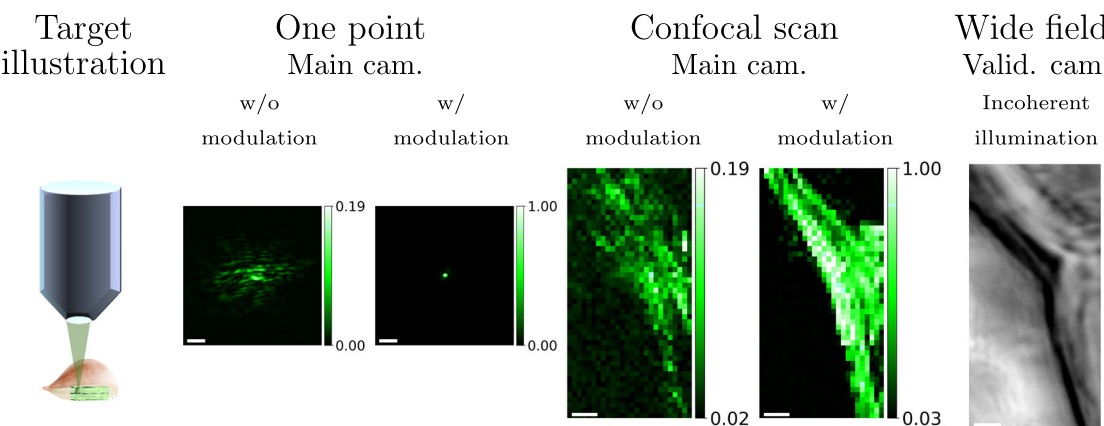

**Fig. 7 | Confocal scan of onion cells.** Our algorithm was used to image onion cells inside a 140 μm thick onion slice. The first column: an illustration of the target. Columns 2–3: Main camera images of a single focal spot before and after correction. Columns 4–5: Confocal scanning results: with and without aberration correction. Column 6: A reference image of the target captured from a validation camera behind the target, under wide-field incoherent illumination. The scale bar is 4 μm.

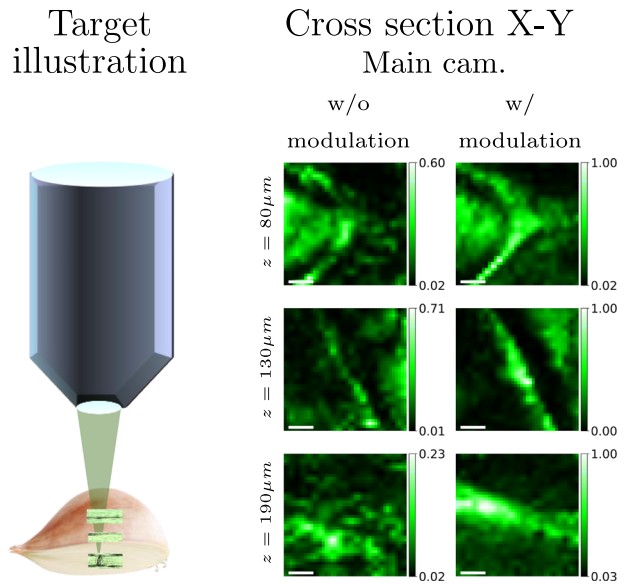

**Fig. 8 | 3D aberration correction of an onion slice.** Our algorithm was used to image onion cells at different depths of 80, 130, and 190 μm. The first column depicts a schematic view of the target. For each layer, we present x-y cross sections. The scale bar on confocal images is 5 μm.

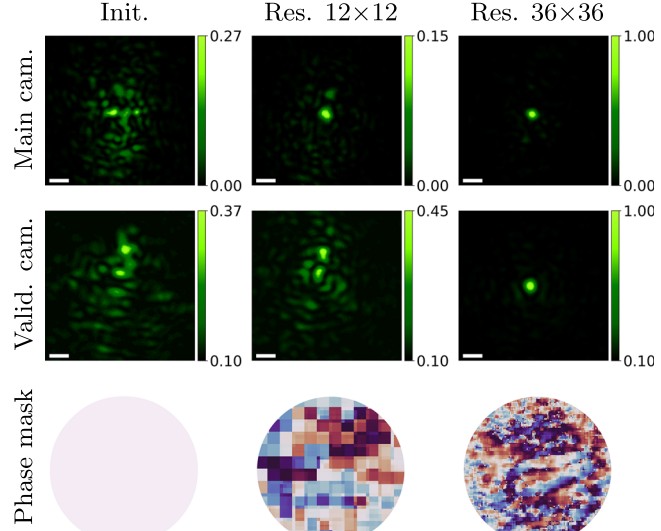

**Fig. 9 | Impact of phase mask resolution.** We evaluate the effectiveness of gradient descent (GD) solutions using phase masks of varying resolutions. We place the chrome mask behind two layers of parafilm, creating a significant aberration. Results demonstrate that higher-resolution phase masks yield superior solutions, enabling convergence in both the main and validation cameras. The scale bar is 4 μm.

development of a fast, aberration-corrected OCT system. Wavefront shaping is also highly effective in correcting the aberrations of fluorescence microscopes. We believe that a modification of our gradient formulation can be adapted to incoherent light sources. However, there are two main challenges for this approach. First, we need to retrieve the emitted complex wavefront from incoherent light sources. This could be tackled by an incoherent interferometer[24,25,65,66]. Second, the low photon count emitted by fluorescent targets makes it challenging to measure the gradient with sufficient SNR.

Our rapid gradient descent optimization algorithm resembles fast time-reversal algorithms[28,31,58–61] for wavefront shaping, which use the power iteration method to estimate the eigenvectors of the reflection matrix. The relationship with these algorithms is explored in detail in the supplementary file. We demonstrate that when the target area includes a single point, our derived gradient is equivalent to a power iteration on the reflection matrix. However, as demonstrated in Fig. 4, for coherent imaging, the largest eigenvector of the reflection matrix

does not always yield an optimal modulation, and adjustments to the power iteration method are not straightforward. In contrast, defining a target score and its gradient provides a principled method for guiding the modulation search towards desired outcomes.

## Methods

### System design

Our setup is fully depicted in Fig. 1 of the supplementary material. A collimated laser beam (CPS532 Thorlabs) illuminates a phase SLM (GAEA 2 Holoeye). The light is then focused with an objective lens (N20X-PF Nikon) on the target plane through the scattering material. The reflected light from the target is collected through the same objective lens and is modulated by a second SLM (LETO Holoeye). Finally, the light is imaged with a camera (Atlas-314S Lucid vision labs). The second SLM (LETO Holoeye) has two functions: create a phase pattern used for estimating the outgoing wavefront via a phase-retrieval algorithm. The second use is to place the conjugate of the

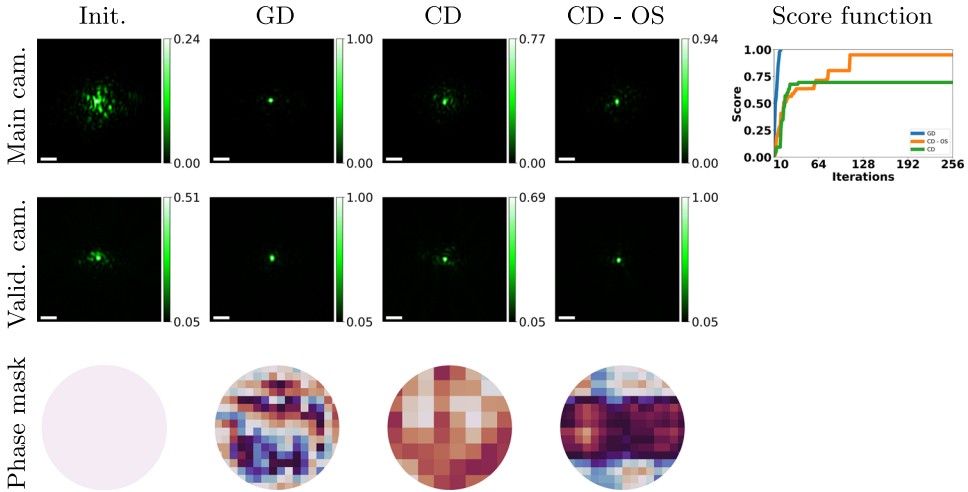

**Fig. 10 | Comparison of gradient descent and coordinate descent schemes.** We compare our gradient descent (GD) optimization to coordinate descent (CD) using the Hadamard basis. Due to CD's time-intensive nature, we optimize for a low-resolution mask and use a weak aberration volume (a single layer of parafilm tape). To address CD's susceptibility to noisy measurements, we test two configurations: In standard CD, we use 5 shots to estimate the phase of each basis element. This approach could not improve the score beyond a certain point, because the improvement in the score is lower than the imaging noise. Next, we use an over-sampled CD (CD-OS), capturing twenty-four images per basis element. This over-sampled version is less susceptible to noise and reached a higher confocal score, at the cost of an even longer acquisition. Overall, our results demonstrate that GD converges significantly faster than both CD variants and exhibits greater robustness to noise. The last column shows the score as function of iterations, where the blue line is GD, the green line is CD, and the orange line is CD-OS. The scale bar is 4 μm.

retrieved phase in order to show confocal scanned images and calculate the score function. In our setup, the SLMs are placed in the Fourier plane of the target. Additional details on the setup, calibration, and alignment can be found in the supplementary material. In Fig. 2 of the supplementary material, we show how to calibrate our system with back illumination.

### Experimental targets

For our targets, we use a highly reflective chrome-coated mask (Nanofilm) and use an in-house lithography process to create patterns with a 2 μm resolution. For scattering material, we used chicken breast tissue with thickness of 130–240 μm or a number of layers of parafilm[63], as stated in the text. For all results of chrome mask target except for the results in Fig. 4, the target area for the algorithm was $\mathscr{A} = 15.6 \mu m \times 15.6 \mu m$ and we scanned points inside this area at twice the Nyquist pitch, with a sampling interval of 1.3 μm. After the algorithm convergence, the final confocal scan was done with a diffraction-limited sampling interval of 0.65 μm. For the target in the fourth row of Fig. 5, the target spans an area larger than $\mathscr{A} = 15.6 \mu m \times 15.6 \mu m$. Hence, we ran the algorithm four times in partially overlapping areas and stitched the results together to form the final image.

For the bead targets, we added quadratic phase in order to focus in different depths, similar to ref.[67]. In Fig. 6, the target volume is $\mathscr{A} = 10.4 \mu m \times 10.4 \mu m \times 14 \mu m$. The sampling interval in lateral axis was 0.5 μm and sampling interval in the axial direction was $2 \mu m$, which is half the diffraction limit.

For onion cell imaging, the target area was $\mathscr{A} = 26 \mu m \times 26 \mu m$ with sampling interval of 1.3 μm. To capture the full image in Fig. 7, we ran the algorithm four times in partially overlapping areas and stitched the results together to form the final image.

In our current setup, the runtime of the algorithm is mainly limited by two components. The first is the liquid-crystal SLM, which works at a rate of approximately 17 Hz for a phase pattern to fully transform. The second is performing gradient acquisition using phase diversity optimization. Each iteration of our algorithm requires acquiring five images for phase diversity optimization for each sampling direction, solving a phase diversity optimization problem for each sampling direction, and performing a gradient step using backtracking line-search (usually a single measurement is needed). For the results in Fig. 10, this means a runtime of 14 min. This is in comparison to CD optimization, which took 900 min in the same setup. By adding a fast SLM (which currently run at 1.4 kHz) and using off-axis interferogram[7,10,13,14,16], runtime could be substantially decreased to mere seconds. In the supplementary material, we further suggest an even faster setup using point-interferometry, and in Fig. 4 in the supplementary material, we present the suggested setup. We believe that with such a point diffraction interferometry implementation, we can estimate a modulation in less than one second.

### Reporting summary

Further information on research design is available in the Nature Portfolio Reporting Summary linked to this article.

## Data availability

The data that support the findings of this study is available in ref. 68.

## Code availability

The code used for acquiring and processing the data is available at ref. 68.

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

## Acknowledgements

This research was funded by ERC SpeckleCorr-101043471, ISF 563/24.

## Author contributions

S.M. designed and constructed the system. S.M. and M.A. performed the experiments. S.M. analyzed the data. A.L. conceived and supervised the project. All authors contributed to writing of the manuscript.

## Competing interests

The authors declare no competing interests.
