## [Transparent Peer Review file · Nature Communications]

Rapid wavefront shaping using an optical gradient acquisition

Corresponding Author: Professor Anat Levin

Version 0:

Reviewer comments:

Reviewer #1

(Remarks to the Author)

This work develops a sample-efficient method for determining the necessary modulation patterns to perform wavefront shaping and focus light over a isoplanatic patch (not just a single point) within a scatterer. The proposed system captures coherent reflection model measurements with one SLM in the illumination path and another in the receive path. An algorithm and methodology is developed to efficiently optimize the pattern projected on those SLMs.

The proposed method works as follows: For a given modulation pattern ρ (what is being optimized), the authors apply additional tilts to steer the beam over a small area (isoplanatic region). The sum of the intensity of the light returned from that scan is used to create a score function---when that score is maximized the SLM will counteract the scattering. The score function and its gradient involve an unknown reflection matrix. The paper's key insight is that the components of the gradient can be optically recorded. This is a very cool insight! Using the gradient measurements, the authors proceed to optimize the score and perform wavefront shaping.

The proposed method is tested samples etched reflective materials and beads imaged through chicken breast and within gel, respectively. The proposed method is shown to be significantly more effective (higher resolution and fewer measurements) than coordinate descent method that optimize hadamard basis coefficient representations of the SLM pattern one-by-one.

With interferometric measurements (not demonstrated) the proposed method could perform high-resolution reflection-mode wavefront shaping with only a handful of measurements. With the current implementation (uses phase diversity measurement to measure field) the system can perform reflection-mode wavefront shaping with only tens of measurements.

Overall, the proposed method is novel, well-validated, and potentially impactful.

Additional Comments and Questions

The current implementation solves a phase retrieval problem (equation (5) in the supplement) to measure the incoming fields? Does this place limitations on the irregularity/resolution of the field that can be recorded? Is (5) still effective if the incoming field is a high-resolution speckle pattern?

Could the proposed method deal with anisoplanatic aberrations with patch-by-patch scanning?

Could the authors comment on the resolution-vs-scattering correction trade-offs associated with the proposed method? It seems one needs to optimize over an extended area on the target in order to avoid simplify optimizing an interference pattern, but a larger area means coarser imaging in a scanned confocal imaging system.

While Fig 2 provides an intuitive explanation, the manuscript would benefit from additional information on why maximizing the score leads to effective wavefront shaping. The paper provides several citations, but one should be able to understand why this score is useful without reading those papers.

L160 states "To address this, we propose averaging the confocal score over a small area, which mitigates these interference

effects." This should be rephrased to clarify whether the "area" refers to; a small area on the camera sensor or the area of interest \mathcal{A} ?

Reviewer #2

(Remarks to the Author)

The manuscript entitled "Rapid wavefront shaping using an optical gradient acquisition" describes gradient descent optimization method for rapid wavefront shaping. Unlike the conventional coordinate descent method, it can update all the control elements of wavefront shaping device having the potential to greatly improve the speed. The authors validate their method with various target structures and scattering medium. In addition, they compared their method with conventional coordinate descent algorithm to prove its application speed.

Strong points:

- New algorithm for gradient descent optimization
- Update all SLM elements at once such that the speed of application is much faster than coordinate descent method
- Flexibility for selecting the score function (in this manuscript, confocal energy within correction area was chosen for the score function.)
- Compared to iterative phase conjugation approaches, this method considers larger area, which enhances the accuracy of the aberration quantification.

Weak points:

- Demonstration level is weak. The authors validate their method with various artificial targets. Nowadays, the wavefront shaping technique is widely used in various adaptive optics devices such as confocal fluorescence, multi-photon microscopy, and optical coherence tomography with real biological tissues.
- The authors validated the method with well defined 2D target structures under artificial scattering medium or chicken breast tissues. However, this is not a natural situation like real biological tissues. For example, is their method works for extended 3D biological structures?
- This method works within an isoplanatic patch area (or within memory effect range). Discussion about the working range and strategy for dealing with multiple isoplanatic patches is necessary.

Technical comments:

- Line 119: How to determine the imaging area? Optimal area size should be related to the isoplanatic patch area. But this is unknown in the initial step. Is there any criteria for selecting the area size?
- Equations (6-7) assume the reciprocity of the scattering medium. In line 193, the authors also assume the reciprocity of the reflection matrix. However, it often fails in real experimental situations due to alignment, limited detection bandwidth, etc. Is this method still valid under the situation?
- Line 205: How to integrate all points l within a single exposure? In line 122, the authors described that they applied tilt-shifted modulation to the SLM for each point l . This means that it requires different SLM phase pattern for different points.

Recommendation: The proposed method presents an interesting approach to wavefront shaping, but its practical applicability to biological imaging is rather unclear. If the authors can demonstrate the feasibility of their method in real biological tissues and address the concerns regarding its limitations in 3D structures and isoplanatic patches, I would recommend the publication of this manuscript.

Reviewer #3

(Remarks to the Author)

In the paper, the authors proposed a wavefront correction set-up for a coherent confocal microscope. The set-up consists of two spatial light modulators, one to correct the illumination beam and one to correct the reflected beam. The authors also proposed a score function to evaluate the quality of the modulation. By optimising the score function using a gradient descent algorithm, the wavefront correction is derived with a runtime of 14 mins. The gradient descent method used five defocused images obtained on a camera to estimate phase from speckle. The main advance that is claimed above previous demonstrations is the application of gradient descent, rather than optimisation on a single coordinate/mode basis. This advance seems to be useful, but it is not on its own significant.

The authors have discussed fairly some limitations of their current approach. For example, one essential term, the reflection matrix R , is unknown and it requires a phase retrieval algorithm to derive the complex field. However, there are still remaining questions about how well the method should work. Note the discussion at the end of 3.1: "a high confocal score at the sensor plane does not guarantee that the light has focused into a single spot inside the tissue, due to various interference effects. To address this, we propose averaging the confocal score over a small area, which mitigates these interference effects". Where is the proof that this works? Is this an assumption? The experimental results indicate that it can work. However, there has been an outstanding question about such "dual-pass" effects in reflection mode/scattering coherent imaging systems for a long time (e.g. since Journal of the Optical Society of America A 12, pp. 195-201 (1995) <https://doi.org/10.1364/JOSAA.12.000195>).

The experimental results presented in the paper do not provide sufficient information about the method's effectiveness in real applications; the realistic effects of signal/noise and other common practical variations were not considered in the

experimental demonstrations. For example, the specimens used in the demonstrations were first chrome masks and then polystyrene beads in agarose, each of which was placed behind a tissue section or parafilm. In both cases, the reflections/scatter from the specimens would be stronger than those obtain purely from inside tissue sections. It is not therefore clear whether this method would work in more realistic systems.

Some things are not clear about the presentation of results in Fig 8: What does “iterations” mean? How many image measurements is an iteration? From the referenced publication (31) the coordinate descent seems to use a single “total signal” to optimise, rather than the more detailed images from a camera used for phase retrieval here. This may not be a fair comparison, as the method in this new paper - in effect - uses a different detection system to the previous method.

Some claims in the paper, such as the method can be “directly applies to other popular imaging schemes such as OCT and fluorescent microscopy” seem non-conclusive as the proposed method was based on a coherent imaging model. While one might imagine this extension could be done, it has not been shown here in this work.

The authors in the paper also proposed a few future potential approaches which, based on them, would have much-improved performances. This would be interesting to see, and possibly useful, but it has again not been implemented in this work.

Based upon these considerations, the paper does not have sufficient significance and is unsuitable for the journal. I would recommend the authors reconsider submitting the paper after they finish their future potential much-improved approaches.

Reviewer #4

(Remarks to the Author)

Version 1:

Reviewer comments:

Reviewer #1

(Remarks to the Author)

I thank the authors for their response and additional experimental results and derivations. All of my concerns have been adequately addressed and I believe the manuscript is fit for publication.

Reviewer #2

(Remarks to the Author)

The authors made a reasonable effort to address most of my concerns. Specifically, they attempted to demonstrate the method's applicability to biological tissues and 3D structures by imaging a volumetric bead distribution and an onion cell layer. However, I remain unconvinced that the presented results are sufficient to substantiate these claims.

First, the resolution of the confocal scan image of the onion cells obtained via SLM modulation appears notably low, despite the claimed focus improvement. The expected sharp boundaries of the cell walls are blurred, and the overall image quality does not convincingly support the method's effectiveness in biological imaging. Furthermore, in contrast to conventional reflection-mode confocal microscopy—where strong signals typically appear along the cell walls—the authors' results exhibit inverted contrast, raising concerns about the sample condition and the accuracy of the imaging system. Clarification of the sample preparation and imaging conditions is needed.

To further validate the method's applicability to biological tissues, it is important to demonstrate volumetric imaging across multiple depths within the onion sample or other representative biological specimens.

In addition, a clearer discussion of how the proposed method differs from existing wavefront shaping approaches is necessary. The method appears to be a hybrid between conventional wavefront sensorless techniques and wavefront sensing schemes: it claims to require fewer iterations than sensorless methods and fewer measurements than full sensing approaches. However, it remains unclear whether this trade-off leads to superior overall performance. Without a more explicit comparison and justification, it is difficult to assess the practical significance of the proposed framework.

A more thorough clarification of these points is essential before I can recommend this manuscript for publication.

Reviewer #3

(Remarks to the Author)

I frame my response around the major points I made in the first review and the corresponding revisions made by the authors:

Major point in original review:

The correction takes 14 minutes, uses five defocussed images. Difference to the previous demonstrations is the application of gradient descent, rather than optimisation on a single coordinate/mode basis. This advance seems to be useful, but it is not on its own significant.

Comment on revisions:

This fundamental concern has not been addressed.

Major point in original review:

Assumptions about dual pass measurements using coherent light – are they valid?

Comment on revisions:

[Authors' revision point 2] I think the authors have misinterpreted the cited paper. They claim it relies on an incoherent imaging model, which looks like a selective interpretation of the content. It uses a coherent model, after which it is shown to behave like an incoherent process, following averaging. There is still an ambiguity, which is readily seen through even a ray optics model for a specular reflection in the focus, whereby odd aberrations cancel out and even aberrations are doubled. Given that the referenced work and the practical results show that the system functions, I am not overly concerned by this point, but it would be good to acknowledge the potential measurement/interpretation issues here.

Major point in original review:

lack of validation in realistic tissue imaging scenarios.

Comment on revisions:

[Authors' revision point 3] Further results have been included using onion cells. This is still a somewhat non-representative model compared to real biological tissue imaging applications. Hence, this point has not been adequately addressed.

Major point in original review:

whether there was a fair comparison between previous and current work in Fig 8.

Comment on revisions:

[Authors' revision point 4] While this response does not really address my point about making a fair comparison in the figure, I can accept this.

Major point in original review:

whether one can assert this is relevant to OCT and fluorescence microscopy

Comment on revisions:

[Authors' revision point 5] Some text has been added to further explain this point, which is a suitable response.

Major point in original review:

"Based upon these considerations, the paper does not have sufficient significance and is unsuitable for the journal. I would recommend the authors reconsider submitting the paper after they finish their future potential much-improved approaches."

Comment on revisions:

While the revisions have responded to some of the points raised in the review and led to the inclusion of more theoretical and experimental results, the fundamental question about the significance of the overall work still stands. The revisions do not change my opinion about this – the paper does not demonstrate sufficient significance for publication in this journal.

Reviewer #4

(Remarks to the Author)

Version 2:

Reviewer comments:

Reviewer #2

(Remarks to the Author)

The authors have provided relatively thorough responses to my queries. While the imaging results for the biological samples and 3D structures may not be of high quality, they are adequate to validate the effectiveness of the proposed method. Given that this work introduces a novel concept and the presented results are sufficient as a proof of concept, I recommend its publication in Nature Communications.

We thank the reviewers for carefully reading our manuscript and for their valuable comments. We have carefully revised the paper following the comments. In particular, we include additional results: 1) To demonstrate the feasibility of our approach on biological samples we now demonstrate imaging inside an onion layer. 2) We were requested to demonstrate how our technique applies for 3D targets, for that we now use transparent beads dispersed inside a volume of agarose gel (where the refractive indices of the gel and beads matches the typical values in tissue), and visualize an aberration corrected image of a 3D sub-volume. Please see below a point by point response to reviewer comments. We also added an additional copy of the paper where the changes were marked in blue.

Reviewer #1 (Remarks to the Author): This work develops a sample-efficient method for determining the necessary modulation patterns to perform wavefront shaping and focus light over a isoplanatic patch (not just a single point) within a scatterer. The proposed system captures coherent reflection model measurements with one SLM in the illumination path and another in the receive path. An algorithm and methodology is developed to efficiently optimize the pattern projected on those SLMs. The proposed method works as follows: For a given modulation pattern ρ (what is being optimized), the authors apply additional tilts to steer the beam over a small area (isoplanatic region). The sum of the intensity of the light returned from that scan is used to create a score function---when that score is maximized the SLM will counteract the scattering. The score function and its gradient involve an unknown reflection matrix.

The paper's key insight is that the components of the gradient can be optically recorded. This is a very cool insight! Using the gradient measurements, the authors proceed to optimize the score and perform wavefront shaping. The proposed method is tested samples etched reflective materials and beads imaged through chicken breast and within gel, respectively. The proposed method is shown to be significantly more effective (higher resolution and fewer measurements) than coordinate descent method that optimize hadamard basis coefficient representations of the SLM pattern one-by-one. With interferometric measurements (not demonstrated) the proposed method could perform high-resolution reflection-mode wavefront shaping with only a handful of measurements. With the current implementation (uses phase diversity measurement to measure field) the system can perform reflection-mode wavefront shaping with only tens of measurements. Overall, the proposed method is novel, well-validated, and potentially impactful.

Additional Comments and Questions

1) The current implementation solves a phase retrieval problem (equation (5) in the supplement) to measure the incoming fields? Does this place limitations on the irregularity/resolution of the field that can be recorded? Is (5) still effective if the incoming field is a high-resolution speckle pattern?

As long as the magnification of the imaging system is set such that the speckle grain is larger than a pixel there is no inherent limitation on the resolution of the recovered modulation, although when the modulation has more parameters the phase retrieval optimization may fail to converge (lines 217-219 in Sec. 3.2. in the supplementary).

2) Could the proposed method deal with anisoplanatic aberrations with patch-by-patch scanning?

Some of the results in the original submission were indeed obtained by stitching together multiple local modulations (The last row of Fig. 5 and Fig. 7). In the revised manuscript we better emphasize this (lines 291-293 and 323-327 in the main paper). We also show the sub-components in supplementary Figs. 8 and 11.

3) Could the authors comment on the resolution-vs-scattering correction trade-offs associated with the proposed method? It seems one needs to optimize over an extended area on the target in order to avoid simplify optimizing an interference pattern, but a larger area means coarser imaging in a scanned confocal imaging system.

The system always measures the confocal intensity at a diffraction limited spot, and the area being modified is not the area of the confocal aperture, but the area of the target we try to correct (\mathcal{A} in Eq. 7). Though, clearly \mathcal{A} is limited by the extent of ME correlation (line 289-293 in the main text).

4) While Fig 2 provides an intuitive explanation, the manuscript would benefit from additional information on why maximizing the score leads to effective wavefront shaping. The paper provides several citations, but one should be able to understand why this score is useful without reading those papers.

We added to the supplementary Sec. 2 a short derivation that explains the score function using the memory effect correlation.

5) L160 states "To address this, we propose averaging the confocal score over a small area, which mitigates these interference effects." This should be rephrased to clarify whether the "area" refers to; a small area on the camera sensor or the area of interest \mathcal{A} ?

Yes, the area here is \mathcal{A} , not area on the camera sensor, we have revised the text (line 179 in the main text).

Reviewer #2 (Remarks to the Author): The manuscript entitled "Rapid wavefront shaping using an optical gradient acquisition" describes gradient descent optimization method for rapid wavefront shaping. Unlike the conventional coordinate descent method, it can update all the control elements of wavefront shaping device having the potential to greatly improve the speed. The authors validate their method with various target structures and scattering medium. In addition, they compared their method with conventional coordinate descent algorithm to prove its application speed.

Strong points:

- New algorithm for gradient descent optimization
- Update all SLM elements at once such that the speed of application is much faster than coordinate descent method
- Flexibility for selecting the score function (in this manuscript, confocal energy within correction area was chosen for the score function.)
- Compared to iterative phase conjugation approaches, this method considers larger area, which enhances the accuracy of the aberration quantification.

Weak points:

- Demonstration level is weak. The authors validate their method with various artificial targets. Nowadays, the wavefront shaping technique is widely used in various adaptive optics devices such as confocal fluorescence, multi-photon microscopy, and optical coherence tomography with real biological tissues.

1) The authors validated the method with well defined 2D target structures under artificial scattering medium or chicken breast tissues. However, this is not a natural situation like real biological tissues. For example, is their method works for extended 3D biological structures?

Note that the bead target of our original manuscript is a 3D target, this is a volumetric dispersion of beads inside gel. We added in Fig. 6 a sketch of the sample to better illustrate what is being imaged. In Fig. 6 in the main paper we now also illustrate the correction of a 3D sub-volume inside this dispersion, rather than only 2D slices. We also added in Fig. 7 a new result of a 3D slice of onion, and we show focusing inside this 3D target.

2) This method works within an isoplanatic patch area (or within memory effect range). Discussion about the working range and strategy for dealing with multiple isoplanatic patches is necessary.

Some of the results in the original submission were indeed obtained by stitching together multiple local modulations (The last row of Fig. 5 and Fig. 7). In the revised manuscript we better emphasize this (lines 291-293 and 323-327 in the main paper). We also show the sub-components in supplementary Figs. 8 and 11.

3) Technical comments: - Line 119: How to determine the imaging area? Optimal area size should be related to the isoplanatic patch area. But this is unknown in the initial step. Is there any criteria for selecting the area size?

Yes, we tried a few isoplanatic area sizes for each sample and kept the ones providing the best results. However, usually once we find a good area for one isoplanatic window we can scan many windows in the same sample using the same \mathcal{A} parameter, and stitch them together (lines 81-90 in the supplementary material).

4) Equations (6-7) assume the reciprocity of the scattering medium. In line 193, the authors also assume the reciprocity of the reflection matrix. However, it often fails in real experimental situations due to alignment, limited detection bandwidth, etc. Is this method still valid under the situation?

We indeed used a careful calibration process to ensure the reciprocity of our system, and to accurately align the detection and the illumination arms. Our system alignment algorithms are detailed in the supplementary file Sec 1.

5) Line 205: How to integrate all points l within a single exposure? In line 122, the authors described that they applied tilt-shifted modulation to the SLM for each point l . This means that it requires different SLM phase pattern for different points.

For this to hold the SLM should not be placed at the Fourier plane, but at a plane conjugated to the aberration plane of the sample, so that there is only a tilt between the modulation and not tilt-shift. The tilt can then be implemented by a fast galvo scanning, and this scanning can happen within the exposure of one camera shot. We explain this in the supplementary file Sec 3.3.

Recommendation: The proposed method presents an interesting approach to wavefront shaping, but its practical applicability to biological imaging is rather unclear. If the authors can demonstrate the feasibility of their method in real biological tissues and address the concerns regarding its limitations in 3D

structures and isoplanatic patches, I would recommend the publication of this manuscript.

The revised manuscript includes an additional biological example. Our targets are 3-dimensional, we now better illustrate the schematic, so this is clearer. We also visualize the correction of a 3D sub-volume rather than only 2D slices.

Reviewer #3 (Remarks to the Author): In the paper, the authors proposed a wavefront correction set-up for a coherent confocal microscope. The set-up consists of two spatial light modulators, one to correct the illumination beam and one to correct the reflected beam. The authors also proposed a score function to evaluate the quality of the modulation. By optimising the score function using a gradient descent algorithm, the wavefront correction is derived with a runtime of 14 mins. The gradient descent method used five defocused images obtained on a camera to estimate phase from speckle. The main advance that is claimed above previous demonstrations is the application of gradient descent, rather than optimisation on a single coordinate/mode basis. This advance seems to be useful, but it is not on its own significant. The authors have discussed fairly some limitations of their current approach. For example, one essential term, the reflection matrix R , is unknown and it requires a phase retrieval algorithm to derive the complex field.

Note that the reflection matrix is unknown but this is a limitation that our approach overcomes elegantly. Despite the fact that R is unknown its operation on any modulation of interest can be measured by the optical system. The phase retrieval process is not designed to measure the full reflection matrix \mathcal{R} , it is only used to estimate the phase of the wavefront $\mathcal{R}u$, where u is the current SLM modulation (lines 212-225 in the main paper).

1) However, there are still remaining questions about how well the method should work. Note the discussion at the end of 3.1: “a high confocal score at the sensor plane does not guarantee that the light has focused into a single spot inside the tissue, due to various interference effects. To address this, we propose averaging the confocal score over a small area, which mitigates these interference effects”. Where is the proof that this works? Is this an assumption? The experimental results indicate that it can work.

This is a well-established model that was exploited by several previous approaches, see Refs. [9,10,14] in the main paper. However, for the shake of completeness the supplementary of the revised manuscript now includes the proof in Sec 2.

2) However, there has been an outstanding question about such “dual-pass” effects in reflection mode/scattering coherent imaging systems for a long time (e.g. since Journal of the Optical Society of America A 12, pp. 195-201(1995) <https://doi.org/10.1364/JOSAA.12.000195>).

Thanks for the reference. After carefully reading it, we noticed that it relies on an **incoherent** image formation model. We agree that in the incoherent case the double-pass system leaves ambiguities and the aberration cannot be fully retrieved. However, this problem is not maintained when using a fully coherent imaging model. While in the incoherent case the aberration is described by the auto-correlation of the positive point spread function, In the coherent case the aberration is described by the auto-convolution (not auto-correlation) of the complex amplitude transfer function. In other words, in the fully coherent case, if $O(w)$ is the complex optical transfer function of one pass through the optical system, the OTF of a double pass through the system is given by $O(w)^2$, not by $|O(w)|^2$, and the phase is not lost.

3) The experimental results presented in the paper do not provide sufficient information about the method’s effectiveness in real applications; the realistic effects of signal/noise and other common practical variations were not considered in the experimental demonstrations. For example, the specimens used in the demonstrations were first chrome masks and then polystyrene beads in agarose, each of which was placed behind a tissue section or parafilm. In both cases, the reflections/scatter from the specimens would be stronger than those obtain purely from inside tissue sections. It is not therefore clear whether this method would work in more realistic systems.

The revised manuscript now includes a biological sample of onion cells (Fig. 7). Following your and reviewer’s 2 comments we better clarified that the target beads were dispersed in agarose gel in a 3D volume where the scattering material were beads from earlier layers and not a slice of chicken breast (Fig. 6 in the main paper and Fig. 5 in the supplementary material). These are transparent polystyrene beads dispersed in agarose gel. The refractive index of these beads is about 1.59, while the refractive index of the gel around them is 1.33. The difference in refractive indices is in the order of magnitude of the differences one finds in biological samples (lines 294-315 in the main paper).

4) Some things are not clear about the presentation of results in Fig 8: What does “iterations” mean? How many image measurements is an iteration? From the referenced publication (31) the coordinate descent seems to use a single “total

signal” to optimise, rather than the more detailed images from a camera used for phase retrieval here. This may not be a fair comparison, as the method in this new paper - in effect - uses a different detection system to the previous method.

It is true that coordinate descent only measures the score, and this score can be measured by a single pixel detector, as opposed to gradient descent which requires capturing a 2D wavefront. While single pixel detectors reach a much higher framerate than 2D sensors, the speed limitation in both cases would be the SLM speed. High-end SLMs operate at 1.4KHz, whereas high end 2D sensors reach more than 2Kfps. For coherent microscope, which we demonstrate, the SNR should also not be a problem as we can increase the laser power (the power that we use for onion cells was less than 2mw).

5) Some claims in the paper, such as the method can be “directly applies to other popular imaging schemes such as OCT and fluorescent microscopy” seem non-conclusive as the proposed method was based on a coherent imaging model. While one might imagine this extension could be done, it has not been shown here in this work. The authors in the paper also proposed a few future potential approaches which, based on them, would have much-improved performances. This would be interesting to see, and possibly useful, but it has again not been implemented in this work. Based upon these considerations, the paper does not have sufficient significance and is unsuitable for the journal. I would recommend the authors reconsider submitting the paper after they finish their future potential much-improved approaches.

As for OCT, since it relies on coherent light, the optimization framework presented is directly applicable. A similar score for OCT imaging was also presented in previous works (Refs. [9,10,14] in the main paper) for digital aberration correction using an equivalent image formation model. Regarding fluorescent microscopy, we add a clarification in the discussion section-- our method could be adapted to fluorescent microscopy; however, we admit that it would require a different approach to measure the wavefront. While we have a plan for how to do that, we agree it is a subject for a different paper (lines 370-384 in the main paper).

Dear Editor and Reviewers,

We thank the Editor and Reviewers for their thoughtful feedback and for the opportunity to submit a second revision of our manuscript. We are encouraged by Reviewer 1's strong endorsement for publication and appreciate the constructive comments provided by Reviewers 2 and 3. Following the Editor's guidance on which points were essential for this revision, we have performed targeted additional experiments and clarifications that address those core issues, as detailed below.

General Statement

Our manuscript introduces a new optical computing–based approach for accelerating wavefront shaping optimization. Traditional algorithms update each modulation parameter sequentially, making them prohibitively slow when many degrees of freedom are involved. In contrast, our method allows the gradient of the cost function to be measured optically, enabling simultaneous updates of all parameters. This represents a conceptual and practical advance that significantly speeds up optimization and opens new directions in adaptive optics for imaging through scattering media.

In the first round of revisions, we conducted extensive additional experimental validation—including biological imaging of an onion slice and volumetric corrections using bead targets. In this second round, following the specific comments from the Reviewers and Editor, we performed further experiments and analyses:

- We investigated and resolved the origin of the inverted contrast observed in onion images.
- We acquired new scans of onion slices under improved conditions, yielding clearer visualization of the cell walls.
- We added onion imaging at multiple depth planes to demonstrate volumetric correction capability.
- We added a comparison with a digital correction approach.

Unfortunately, we have exhausted our resources and have been unable to obtain further suitable biological samples. The main obstacle is that our current first prototype uses a reflection-mode coherent confocal microscope as an imaging system. This system is

inherently limited in biological imaging due to weak back-reflection and the mixing of reflections from different depth layers (lack of optical sectioning). We believe that the full potential of our algorithm will be best realized in future work combining it with OCT, which offers depth sectioning and suffers from similar scattering aberrations that our method can correct. However, building such a system is a multi-year endeavor and beyond the scope of this manuscript. We added an explanation of this limitation to the discussion section.

While we recognize the reviewers' wish for additional biological demonstrations, we believe that the expectations exceed what can reasonably be achieved within a single paper. Our work demonstrates both theoretical novelty and substantial experimental validation—meeting and exceeding the level of demonstration typical of recent high-impact publications in this field. For comparison, we summarize several representative works:

- [1] Haim, O., Boger-Lombard, J., Katz, O.: *Image-guided computational holographic wavefront shaping*. *Nature Photonics* (2024). -- Used a thin layer of onion placed at a large distance behind a commercial glass diffuser.
- [2] Mididoddi, C.K., Kilpatrick, R.J., Sharp, C., Hougne, P., Horsley, S.A.R., Phillips, D.B.: *Threading light through dynamic complex media*. *Nature Photonics* 19, 123 - 134 (2025). -- Scattering medium was only simulated using a spatial light modulator.
- [3] Baek, Y., Aguiar, H.B., Gigan, S.: *Phase conjugation with spatially incoherent light in complex media*. *Nature Photonics* 17(12), 1114 -1119 (2023). -- A very sparse set of fluorescent beads placed behind a glass diffuser.
- [4] Oh, C., Hugonnet, H., Lee, M. et al. *Digital aberration correction for enhanced thick tissue imaging exploiting aberration matrix and tilt-tilt correlation from the optical memory effect*. *Nat Commun* 16, 1685 (2025). --Scattering medium was a polymer coated coverslip.
- [5] Cui, M., Kahraman, S.S. & Wang, L.V. *Optical focusing into scattering media via iterative time reversal guided by absorption nonlinearity*. *Nat Commun* 16, 7807 (2025). --- Imaged extended film made of eosin and gelatin sandwiched between two glass diffusers.

- [6] Feng, B.Y., Guo, H., Xie, M., Boominathan, V., Sharma, M.K., Veeraraghavan, A., Metzler, C.A.: *Neuws: Neural wavefront shaping for guidestar-free imaging through static and dynamic scattering media. Science Advances* 9(26), (2023). -- Various prepared microscope targets slides, placed behind independent diffusers made of onion skin, a commercial glass diffuser, and a glass slide coated with dried nail polish.
- [7] Weinberg, G., Sunray, E., Katz, O.: *Noninvasive megapixel fluorescence microscopy through scattering layers by a virtual incoherent reflection matrix. Science Advances* 10(47), 5218 (2024).-- Only used fluorescent beads or pollens placed at a large distance behind a glass diffuser or a slice of chicken breast tissue.

In comparison, our manuscript tackles a far more challenging and realistic configuration where the target is **embedded within** the scattering medium—leading to spatially varying aberrations. Such volumetric conditions are rarely addressed in previous work.

If the editorial decision is that further biological demonstrations are essential for publication, we will of course respect that determination and resubmit our manuscript elsewhere. However, we hope that our revised results, clarifications, and the broader context above will allow the Editor and Reviewers to appreciate that the manuscript now provides a rigorous and impactful contribution that meets *Nature Communications* standards.

Detailed Point-by-Point Responses

Reviewer #1 (Remarks to the Author):

I thank the authors for their response and additional experimental results and derivations. All of my concerns have been adequately addressed and I believe the manuscript is fit for publication.

We sincerely thank the reviewer for this positive and supportive assessment.

Reviewer #2 (Remarks to the Author):

The authors made a reasonable effort to address most of my concerns. Specifically, they attempted to demonstrate the method's applicability to biological tissues and 3D structures by imaging a volumetric bead distribution and an onion cell layer. However, I remain unconvinced that the presented results are sufficient to substantiate these claims.

First, the resolution of the confocal scan image of the onion cells obtained via SLM modulation appears notably low, despite the claimed focus improvement. The expected sharp boundaries of the cell walls are blurred, and the overall image quality does not convincingly support the method's effectiveness in biological imaging. Furthermore, in contrast to conventional reflection-mode confocal microscopy—where strong signals typically appear along the cell walls—the authors' results exhibit inverted contrast, raising concerns about the sample condition and the accuracy of the imaging system. Clarification of the sample preparation and imaging conditions is needed.

We thank the reviewer for his comment. The reviewer is right that the onion appeared to exhibit inverted contrast. We performed additional experiments to study this. In the previous results, we placed the onion inside water-based gel, so that it would not dry during imaging. Apparently, this gel had a different salt concentration, which was higher in the gel causing osmosis where water is extracted from the cell to equalize concentration of the salt. This resulted in a process known as plasmolysis, where the onion cell shrinks.

We have repeated the experiment without the gel, resulting in the expected bright reflections from the cell walls. The previous result was removed, and the new images (Fig. 7) show stronger contrast consistent with standard reflection-mode confocal imaging.

To further validate the method's applicability to biological tissues, it is important to demonstrate volumetric imaging across multiple depths within the onion sample or other representative biological specimens.

As suggested, we added Fig. 8 demonstrating corrected images at multiple depth slices of the onion sample, showing that the method achieves depth-resolved aberration correction.

In addition, a clearer discussion of how the proposed method differs from existing wavefront shaping approaches is necessary. The method appears to be a hybrid between conventional wavefront sensorless techniques and wavefront sensing schemes: it claims to require fewer iterations than sensorless methods and fewer measurements than full sensing approaches. However, it remains unclear whether this trade-off leads to superior overall performance. Without a more explicit comparison and justification, it is difficult to assess the practical significance of the proposed framework.

We thank the reviewer for this thoughtful comment. Sensorless optimization methods typically require many iterations, whereas our optical-gradient approach converges with significantly fewer iterations, as shown in Fig. 10.

Regarding direct wavefront-sensing approaches, there are two main categories:

1. **Guide-star-based methods**, which directly measure the aberration of a localized beacon. These are not applicable in our case, as no guide-star is available—we instead optimize a score function defined on the measured intensity.
2. **Interferometric approaches** that capture the full complex reflection matrix and fit a parametric model of the aberration. While these share conceptual similarities with our method, there are important distinctions:
 - The correction in these methods is applied digitally after numerical reconstruction, whereas our correction is implemented optically through the SLM. Because reflection-matrix reconstructions are often noisy, this digital correction is susceptible to error propagation. In contrast, applying the correction optically yields a higher signal-to-noise ratio, which is particularly beneficial when imaging weak fluorescent targets or in OCT for deep-tissue structures.
 - Our approach involves a shorter scan. Reflection-matrix imaging typically requires scanning a large area that covers the speckle support around each corrected point, while our method scans only the local region of interest.
 - Reflection-matrix methods involve intensive computational reconstruction,

whereas our approach leverages the optics themselves to perform part of this computation in parallel.

To illustrate the practical difference between optical and digital correction, we have added a brief comparison in supplementary file, Fig. 13. Using previously acquired data, we applied an online implementation of the **CLASS** algorithm (Choi *et al.*) to the scattered wavefronts measured with our system. When applied to data from the first iteration, the CLASS reconstruction is notably noisy. However, after placing our algorithm's correction on the SLM and re-measuring the wavefronts, the CLASS reconstruction quality improves markedly, confirming that optical pre-correction enhances subsequent digital analyses.

An interesting future direction would be to combine these two approaches—using the scattered wavefronts measured in the first iteration to fit an aberration model via multiple gradient-descent iterations, then placing this model on the SLM and re-measuring data under reduced scattering. Such a hybrid strategy could reduce the number of capture iterations required, although, given that our algorithm typically converges within about ten iterations, the expected gain is modest.

A more thorough clarification of these points is essential before I can recommend this manuscript for publication.

Reviewer #3 (Remarks to the Author):

I frame my response around the major points I made in the first review and the corresponding revisions made by the authors:

Major point in original review:

The correction takes 14 minutes, uses five defocussed images. Difference to the previous demonstrations is the application of gradient descent, rather than optimisation on a single coordinate/mode basis. This advance seems to be useful, but it is not on its own significant.

Comment on revisions:

This fundamental concern has not been addressed.

We acknowledge that our current prototype is limited by the refresh rate of the SLM and by the use of phase diversity for phase recovery. These are engineering, not conceptual, constraints. Using faster hardware or existing acceleration schemes would directly improve speed without altering the principle. The conceptual novelty—optically measured gradients enabling simultaneous updates—remains robustly demonstrated.

Major point in original review:

Assumptions about dual pass measurements using coherent light – are they valid?

Comment on revisions:

[Authors' revision point 2] I think the authors have misinterpreted the cited paper. They claim it relies on an incoherent imaging model, which looks like a selective interpretation of the content. It uses a coherent model, after which it is shown to behave like an incoherent process, following averaging. There is still an ambiguity, which is readily seen through even a ray optics model for a specular reflection in the focus, whereby odd aberrations cancel out and even aberrations are doubled.

Given that the referenced work and the practical results show that the system functions, I am not overly concerned by this point, but it would be good to acknowledge the potential measurement/interpretation issues here.

We agree that the paper starts with a coherent model, but indeed states that it behaves as an incoherent model due to averaging. This averaging seems to be the main difference between the paper model and our system. Our system images the sample in a

magnification at which individual spackle grains are visible, no such averaging is involved. Therefore, we can utilize the additional benefits of the coherent model.

Another key distinction relative to the referenced work lies in the type of information captured. In their analysis, only intensity information is measured, which limits reconstruction to the modulation transfer function (MTF). Consequently, their estimated optical transfer function (OTF) exhibits a constant phase transfer function (PTF) in the double-pass configuration. In the concluding remarks of their paper, the authors explicitly acknowledge this limitation and note that recovering the PTF requires access to the phase information. To obtain this missing information, a phase-retrieval algorithm must be employed—precisely the approach implemented in our work.

Major point in original review:

lack of validation in realistic tissue imaging scenarios.

Comment on revisions:

[Authors' revision point 3] Further results have been included using onion cells. This is still a somewhat non-representative model compared to real biological tissue imaging applications. Hence, this point has not been adequately addressed.

Major point in original review:

whether there was a fair comparison between previous and current work in Fig 8.

Comment on revisions:

[Authors' revision point 4] While this response does not really address my point about making a fair comparison in the figure, I can accept this.

Major point in original review:

whether one can assert this is relevant to OCT and fluorescence microscopy

Comment on revisions:

[Authors' revision point 5] Some text has been added to further explain this point, which is a suitable response.

Major point in original review:

“Based upon these considerations, the paper does not have sufficient significance and is unsuitable for the journal. I would recommend the authors reconsider submitting the paper after they finish their future potential much-improved approaches.”

Comment on revisions:

While the revisions have responded to some of the points raised in the review and led to the inclusion of more theoretical and experimental results, the fundamental question about the significance of the overall work still stands. The revisions do not change my opinion about this – the paper does not demonstrate sufficient significance for publication in this journal.

We respectfully note that our approach introduces a fundamentally new way to accelerate wavefront shaping via optical computation—a contribution that can impact multiple imaging modalities, including future OCT and fluorescence systems. The added 3D demonstrations further strengthen this conclusion.

Reviewer #4 (Remarks to the Author):

Closing Statement

We hope the Editor and Reviewers will find that this revision satisfactorily addresses all essential comments. The manuscript now includes clearer explanations, and additional experiments. We believe it meets the criteria for publication in Nature Communications and will serve as a valuable contribution to the optical imaging community.

Sincerely,

Sagi Monin, Marina Alterman, Anat Levin